# Lipid Droplets: Formation, Degradation, and Their Role in Cellular Responses to Flavivirus Infections

**DOI:** 10.3390/microorganisms12040647

**Published:** 2024-03-24

**Authors:** James Z. Hsia, Dongxiao Liu, LaPrecious Haynes, Ruth Cruz-Cosme, Qiyi Tang

**Affiliations:** Department of Microbiology, Howard University College of Medicine, Washington, DC 20059, USA; james.hsia@bison.howard.edu (J.Z.H.); dongxiao.liu@bison.howard.edu (D.L.); laprecious.haynes@bison.howard.edu (L.H.); ruth.cruzcosme@howard.edu (R.C.-C.)

**Keywords:** lipid droplet, antiviral, flaviviruses, orthoflaviviruses, dengue virus, Zika virus, hepatitis C virus, lipid droplet contact, lipophagy, lipid droplet biogenesis, endoplasmic reticulum, fatty acid

## Abstract

Lipid droplets (LDs) are cellular organelles derived from the endoplasmic reticulum (ER), serving as lipid storage sites crucial for maintaining cellular lipid homeostasis. Recent attention has been drawn to their roles in viral replication and their interactions with viruses. However, the precise biological functions of LDs in viral replication and pathogenesis remain incompletely understood. To elucidate the interaction between LDs and viruses, it is imperative to comprehend the biogenesis of LDs and their dynamic interactions with other organelles. In this review, we explore the intricate pathways involved in LD biogenies within the cytoplasm, encompassing the uptake of fatty acid from nutrients facilitated by CD36-mediated membranous protein (FABP/FATP)-FA complexes, and FA synthesis via glycolysis in the cytoplasm and the TCL cycle in mitochondria. While LD biogenesis primarily occurs in the ER, matured LDs are intricately linked to multiple organelles. Viral infections can lead to diverse consequences in terms of LD status within cells post-infection, potentially involving the breakdown of LDs through the activation of lipophagy. However, the exact mechanisms underlying LD destruction or accumulation by viruses remain elusive. The significance of LDs in viral replication renders them effective targets for developing broad-spectrum antivirals. Moreover, considering that reducing neutral lipids in LDs is a strategy for anti-obesity treatment, LD depletion may not pose harm to cells. This presents LDs as promising antiviral targets for developing therapeutics that are minimally or non-toxic to the host.

## 1. Introduction

Flaviviridae is a virus family consisting of three primary viral genera: orthoflaviviruses (formerly flaviviruses), hepaciviruses, and pestiviruses [1]. These viruses are characterized by an enveloped viral particle containing a 10–11 kb single-stranded positive RNA genome [2]. Among them, pestiviruses pose a significant threat, causing viral diseases in mammals like cattle, sheep, goat, and swine. In the context of human pathogens in the Flaviviridae family, orthoflaviviruses and hepaciviruses take precedence. Orthoflaviviruses encompass West Nile virus, dengue virus (DENV), tick-borne encephalitis virus, yellow fever virus, Zika virus (ZIKV), and several other viruses which may cause encephalitis [3]. Orthoflaviviruses typically transmit through mosquitos or ticks, while hepaciviruses, with 14 members, are exemplified by hepatitis C virus (HCV), causing hepatitis in humans and transmitted through bloodborne routes like blood transfusion or injection. The infection mechanisms of flaviviruses involve interactions with cellular receptors.

One of the critical interactions between an orthoflavivirus and its permissive and infected cells involves the indirect or direct effects of the virus on cellular lipid components, which represent a dual impact [4,5,6]. Lipid-mediated viral interaction with host cells enables viruses, enveloped viruses in particular, to enter permissive cells through interaction with the cell membrane. On the other hand, the cellular lipid components may play a role as a sensor of invaders, signaling the cells to initiate their defensive mechanisms. Once the viruses break through the cell membrane, they will fight with and harness the cellular lipid in the whole viral replication cycle, including viral growth, assembly, and budding to release from cells. Viral interactions with cellular lipids may occur throughout the viral replication cycle starting from viral entry and ending with viral egress, preparing the virus to initiate a new replication cycle in another cell.

Two important and microscopically visualizable lipid structures, the lipid raft and lipid droplet (LD), have recently garnered attention in the research community. Lipid rafts are small, cholera toxin B (CTB)-bound structures that are highly dynamic [7]. They are the plasma membrane microdomains enriched in cholesterol, glycosphingolipids, and phospholipids. Since they contain receptors for viral infection and generate cellular signaling for viral egress, lipid rafts are utilized by viruses to enter cells by caveolin-mediated endocytosis [6,8]. However, it is noteworthy that lipid rafts also house pathogen-sensing receptors that trigger downstream signaling events, such as programmed cell death or cytokine production for viral clearance [6,9], so lipid rafts may also be antiviral. Upon entry into cells, viruses engage with various cellular organelles to ensure successful replication, one of which is a lipid-rich structure known as the LD.

The LD is a subcellular organelle found dynamically in the cytosol and identified in various eukaryotes, ranging from yeast, insects, plants, and animals to humans [10,11]. LDs typically exhibit a round shape with sizes varying from 20 nm to 100 μm, comprising a monolayer phospholipid membrane and a neutral lipid core [10]. The protective proteins embedded in the monolayer membrane belong to the perilipin family, including perilipin 1, adipophilin (ADRP, also known as perilipin 2), TIP47 (perilipin 3), S3-12 (perilipin 4), and OXPAT/MLDP (perilipin 5) [12]. The lipid core contains neural lipids in the form of triacylglycerol (TAG), cholesteryl esters (CE), and retinyl esters [10,13]. Perilipin proteins shield the lipid core from lipolysis by adipose triglyceride lipase (ATGL), hormone-sensitive lipase (HSL), and triglyceride lipase (TGL) [12]. LDs are a significant component of adipocytes, reaching sizes of up to 100 μm. Hepatocytes also have abundant LDs. In other types of cells, LD sizes typically range from 20 to 40 nm. LDs are dynamically synthesized and broken down in response to cellular needs and environmental signals [14]. However, its biological functions are not yet fully understood. Recent studies have reported interactions between LDs and invaded viruses within cells. This review will specifically concentrate on the viral effects on LDs and the reciprocal impact of LDs on viruses.

## 2. LD Biogenesis and Breakdown

### 2.1. LD Biogenesis

#### 2.1.1. LD Nucleation in Endoplasmic Reticulum (ER)

The biogenesis of LDs involves an enzyme-driven lipid anabolism initiated by the CD36-mediated uptake of fatty acids (FAs) from nutrition [15,16,17] under fed conditions; this process leads to the synthesis of TAG within the two layers of the membrane of the ER [18,19,20]. The uptake of fatty acids from nutrition also requires plasma membrane-bound proteins such as FA binding proteins (FABPs) and fatty acid transport proteins (FATPs) [21]. The FAs can also be synthesized from the glycolysis pathway inside of cells (blue box of Figure 1), the tricarboxylic acid (TCA) cycle in the mitochondrion (pink oval of Figure 1), and FA formation in the cytosol from citrate. The formation of LDs encompasses three biological processes in cells: the synthesis of neutral lipids, including TAG and CE, in the ER, the nucleation of neutral lipid and perilipin proteins in the ER, and the release of LDs from the ER to the cytosol (Figure 1). The synthesis of TAG from glycerol-3-phospholipid (G-3-P) involves four steps following the Kennedy pathway (purple box of Figure 1) [22,23].

Lysophosphatidic Acid (LPA) Synthesis: The initial step is the conversion of glycerol-3-phosphate (G-3-P) to LPA, catalyzed by glycerol-3-phosphate acyltransferases (GPATs).Phosphatidic Acid (PA) Formation: LPA is further acylated to form phosphatidic acid (PA) by 1-acyl-sn-glycerol-3-phosphate O-acyltransferases (AGPATs).Diacylglycerol (DAG) Synthesis: The third step involves the conversion of PA to diacylglycerol (DAG) by Mg^2+^-dependent PA phosphatases (PAP1s).TAG Biosynthesis: The final step is the biosynthesis of TAG from DAG by diacylglycerol O-acyltransferases (DGATs).

The other LD neutral lipid component is CE/SE, which can be derived from low-density lipoprotein (LDL)-derived cholesterol or from nutrients. The sensing of CE is conducted through the interaction of SCAP (SREBP1 cleavage-activating protein) and SREBP (sterol regulatory element-binding protein) [17,24,25]. LDL-derived cholesterol is released into lysosomes and transported to the plasma membrane (PM) and the ER membrane where it is esterified with an FA by Acetyl-Coenzyme A acetyltransferase (ACAT) for storage as CE/SE in LDs. The neutral lipids, TAG and CE/SE, along with perilipin proteins, are nucleated within the ER, forming round-shaped LDs that are subsequently released to the cytosol (Figure 1). Therefore, LDs are derived from the ER, where TAG and CE/SE are synthesized between the two leaflets of the ER membrane [26,27,28].

#### 2.1.2. Cellular Proteins Involved in LD Formation

In addition to the enzymes involved in lipid anabolism for generating TAG and CE/SE in the ER (Figure 1), several cytoplasmic proteins associate with LD formation, budding, and maturation [26].

(1)Fat storage-inducing transmembrane proteins 1 and 2 (FIT1 and FIT2): These are ER proteins, not directly related to TAG biosynthesis, that play crucial roles in TAG/LD formation [29] and LD budding from the ER [30]. FIT proteins, possessing six transmembrane domains, directly bind to TAG, a crucial interaction for their roles in the formation of LDs [31]. Interestingly, Chen et al. revealed that the FIT2 protein interacts with ER tubule-forming proteins and ER skeletal protein septins, influencing the maturation of LDs [32].(2)Seipin (encoded by the BSCL2 gene) [33]: This ER protein is essential for nascent LDs to egress from the ER. Cells with defective seipin retain small LDs within the ER [34], indicating its role in LD budding and maturation [35,36]. Functional overlapping with FIT has been suggested, and further studies are needed to detail the interactions between the two proteins in the ER.(3)Lipin: Positioned in the cytoplasm, lipin is a phosphatidate phosphatase [37]. The mutation or deletion of lipin results in massive adipose tissue loss [38]. However, the specific step of LD formation affected by lipin remains elusive and requires further investigation.

#### 2.1.3. LD-Related Proteins (LDRPs)

LDs in the cytoplasm are enveloped by a monolayer phospholipid membrane and adorned with LDRPs. LDRPs include both integral and peripheral proteins that surround LDs [39]. Peripheral LDRPs maintain a distance from the hydrophobic core. Functionally, LDRPs are engaged in several cellular processes, including lipid metabolism and transport, intracellular trafficking, signaling, chaperone functions, RNA metabolism, and cytoskeletal organization. Physiologically, LDs are highly dynamic organelles, undergoing cycles of growth and consumption that closely parallel cellular metabolism and nutrient availability. Stored lipids are mobilized for energy production during starvation or for phospholipid synthesis during periods of high demand for membrane formation.

LDRPs are categorized into two classes [26]:

Class I LDRPs: These are ER proteins with different cellular functions. Some participate in lipid biosynthesis, such as ACSL3, GPAT4, and DHAT2, while others are involved in the ubiquitin-dependent proteolysis, like ancient ubiquitous protein 1 (AUP1) and UBX domain-containing protein 8 (UBXD8; also known as FAF2). A common characteristic of these proteins is the presence of hydrophobic hairpin domains in the middle, allowing them to insert into the inner side of the LD monolayer envelope.

Class II LDRPs: These proteins are recruited to LDs from the cytosol and bind to LDs through amphipathic α-helices, as observed in CCTa (CTP:phosphocholine cytidylyltransferase) and PLIN4.

LD proteomics have identified a plethora of LDRPs that are classified into different categories based on their functions. Well-demonstrated LDRPs encompass (a) perilipin family proteins, including perilipin 1, adipophilin (ADRP, also known as perilipin 2), TIP47 (perilipin 3), S3-12 (perilipin 4), and OXPAT/MLDP (perilipin 5) [12]; (b) proteins involved in lipid and energy metabolism like HSC70, ATGL, HSL, and ACSL1-3; (c) signaling proteins such as Calcium-binding protein p22, Caveolin1, and Methyltransferase-like protein 7A; and (d) membrane-trafficking proteins like Rab18 and vimentin [40,41,42,43,44]. These proteins encompass those that protect LDs and others involved in degrading the core lipids. Thus, LDRPs play a crucial role in maintaining the homeostasis of LDs throughout the cell’s life.

### 2.2. LD Breakdown

#### 2.2.1. Lipolysis of LD

The homeostasis of LDs in a cell is physiologically regulated through protections offered by certain LDRPs and enzymatic hydrolysis mediated by lipases, a process known as “lipolysis” (Figure 2) or through a selective form of autophagy (lipophagy) (Figure 3) [45,46,47]. Lipolysis involves the sequential and enzyme-driven catabolism of lipids, enabling the hydrolysis of ester bonds between long-chain FAs and the glycerol backbone which is a highly regulated release of FAs. During the initial phase of lipolysis, protein kinase A (PKA) phosphorylates a member of the PAT protein family, PLIN1, leading to its proteasomal degradation [48,49]. This event liberates the ATGL activator protein CGI-58, initiating TAG breakdown through activated ATGL [49,50]. ATGL selectively catalyzes the first step of TAG hydrolysis, producing DAGs and free FAs [51]. Three lipolysis enzymes orderly catalyze three steps of lipolysis: (1) ATGL is responsible for breaking down TAGs to DAG, (2) DAG is then degraded to monoacylglycerol (MAG) by HSL, and (3) MAG is finally broken down into free FAs by monoacylglycerol lipase (MAGL) [51,52,53,54] (Figure 2). This highly regulated process ensures the controlled release of FAs.

#### 2.2.2. Lipophagy of LD

Autophagy stands out as the primary mechanism employed by eukaryotic cells to degrade larger cellular structures, encompassing protein aggregates, damaged organelles, and intracellular pathogens [55]. Lipophagy, a specific form of macroautophagy, has been observed in the starved liver of mice, where Microtubule-Associated Proteins 1A and 1B, Light Chain 3 (LC3) along with Plin1, and ADRP, were detected in both LDs and lysosomes [56,57]. Lipophagy involves the engulfment of LDs within the membrane of the autophagosome, followed by fusion with lysosomes [58,59].

The process of lipophagy unfolds in five stages (Figure 3):(1)Preautophagosome assembly: involves cargo (LDs) selection and packaging.(2)Phagophore membrane expansion: envelops LDs.(3)Vesicle elongation: facilitates LD encapsulation.(4)Vesicle nucleation: forms a closed autophagosome.(5)The fusion of matured autophagosomes with the lysosome. Results in the degradation of engulfed LDs by hydrolytic enzymes are delivered by the lysosome.

In addition to macroautophagy, chaperone-mediated autophagy (CMA) may play a role in the degradation of LDs as shown in Figure 3. CMA-mediated LD degradation depends on interactions between LD protein HSC70 with lysosome protein LAMP-2A. Figure 3 also illustrates that the phosphorylation of Plin2 and Plin3 by AMPK could be involved in the activation of CMA [60]. This multifaceted autophagic landscape, encompassing both macroautophagy and CMA, underscores the intricate regulatory mechanisms employed by cells to maintain LD homeostasis.

To facilitate the breakdown of LDs, lipases from the lipolysis pathway and hydrolytic enzymes participating in lipophagy must penetrate the LD core to access TAGs and CEs. This necessitates the disruption of the monolayer membrane by interfering with perilipin proteins that provide structural support to the LD monolayer membrane. Key proteins implicated in activating lipophagy include peroxisome proliferator-activated receptor β/δ (PPARβ/δ), Ca^2+^/calmodulin-dependent protein kinase kinase β (CaMKKβ), AMP-activated protein kinase (AMPK), Unc-51-like kinase 1 (ULK-1), LC3I/II, and p62 (SQSTM1) [61]. Enzymes involved in lipolysis, such as ATGL and HSL, are activated by CGI-58. Disruptions in the activities of these lipases often lead to abnormalities in humans, as comprehensively reviewed by Onal et al. [44].

## 3. Flaviviruses and LD

### 3.1. Viral Replication Cycle

The replication cycle of orthoflaviviruses (OFVs) begins with their contact with the surface of permissive cells. The surface of flaviviruses, including orthoflaviviruses, is characterized by envelope (E) proteins crucial for attaching to the host cell membrane during internalization via endocytosis [2,62]. The attachment depends on the interaction between the viral E protein and specific receptors. While AXL, Tyro3, TIM1, and DC-sign have been identified as receptors for ZIKV to infect permissive cells [63,64], further investigation is needed to reach a conclusive statement due to the disputed results [65,66]. Viral entry is completed through endocytosis, leading to the release of viral RNA into the ER as the capsid breaks apart in the cytoplasm. The positive-sense RNA genome is then translated by host ribosomes attached to the endoplasmic reticulum, resulting in a polyprotein that undergoes proteolytic processing to yield structural and non-structural proteins. Once all components are correctly assembled, the virion is transported out of the cell via endosomal sorting complexes through the endoplasmic-reticulum–Golgi intermediate compartment (ERGIC) to the Golgi apparatus (Figure 4). Exocytosis is employed for the exit of mature virions from the cell, allowing for a single flavivirus to rapidly amplify its viral load, effectively evading the host immune defense system. This intricate process enables the efficient replication and propagation of flaviviruses within the host organism.

### 3.2. Flaviviruses and LD

Members of the Flaviviridae family are all obligate cellular parasites, relying solely on their permissive host cells to support the entire viral replicative cycle. Infected viruses, including Flaviviridae, extensively utilize lipids as essential components. LDs (LDs), recognized as prevalent cytosolic fat storage organelles, are known to contribute to cellular metabolism beyond mere fat storage. The multifaceted roles of LDs involve intricate interactions with other organelles, as extensively reviewed by Olzmann et al. [26]. As subcellular entities, LDs exhibit direct responsiveness to metabolic stresses such as starvation, nutrient stimulation, and environmental cues, leading to alterations in their composition, number, size, and distribution within cells [67]. The AMP-activated protein kinase (AMPK) signal pathway, central to the biogenesis and breakdown of LDs, plays a pivotal role in orchestrating these dynamic responses. Given the significant impact of LDs on cellular metabolism and their central role in responding to metabolic stresses, the molecular mechanisms governing interactions between LDs and viruses have emerged as a focal point in both cellular biology and virology research. Understanding these interactions provides valuable insights into the intricate interplay between viruses and host cellular processes.

### 3.3. LD and Viruses in General

The replication, gene expression, particle assembly, and maturation of viruses involve intricate biological and chemical reactions within infected cells, all of which require energy in the form of ATP. Flaviviruses, being enveloped, particularly necessitate a significant amount of lipid for the formation of viral envelopes. In this context, LDs play a pivotal role as they can supply both ATP and lipids, making them a crucial target for viral interactions post-infection. The relationship between LDs and RNA viruses, including Flaviviruses, encompasses several steps of interactions. Although the specific details may vary among different RNA viruses, a generalized mechanism can be outlined:

**(1) LD Association:** Upon entry into host cells, RNA viruses, particularly those with positive-sense genomes, establish close associations with LDs [5]. This interaction often involves viral proteins interacting with LD-associated proteins, facilitating a connection between the virus and the LDs. LDs directly interact with other organelles, namely LD contacts [68,69,70], to supply necessary lipids for energy production, membrane biogenesis, and intracellular vesicle trafficking. LDs also serve as important regulators to prevent lipotoxicity and maintain lipid homeostasis.

**(2) LD Remodeling:** Once associated with LDs, viral replication complexes undergo rearrangement and remodeling [71]. This process is essential for creating an environment conducive to efficient viral RNA synthesis and replication. Alterations in LD structure and composition, including changes in the distribution and abundance of LD-associated proteins, play a role in this remodeling.

**(3) Energy and Lipid Utilization:** LDs serve as a dual resource for RNA viruses by providing both ATP and lipids [11]. The energy derived from LDs is crucial for the energy-demanding processes of viral replication, gene expression, particle assembly, and maturation. Simultaneously, the lipids from LDs are utilized for the formation of viral envelopes, particularly important for enveloped viruses like Flaviviruses.

**(4) Replication Complex Assembly:** Within LD-associated viral replication complexes, viral RNA-dependent RNA polymerase (RdRp) enzymes facilitate the replication of the viral genome [72]. The availability of FAs from LDs, among other host cellular components, contributes to the assembly and functioning of these replication complexes.

**(5) Membrane Formation:** As viral replication progresses, newly synthesized viral RNA molecules are encapsulated within membranous structures. These structures, derived from LDs or other cellular membranes, serve as protective compartments for viral RNA replication. The association with LDs helps shield viral RNA molecules from host immune responses [73].

**(6) Viral Assembly and Egress:** Following the replication and packaging of viral RNA, the assembly of viral proteins and the formation of progeny virions occur. Some RNA viruses, like flaviviruses, obtain their envelope proteins directly from LDs or LD-associated membranes during assembly [74]. Once assembled, the progeny virions are released from infected cells, ready to infect new host cells and perpetuate the viral replication cycle.

It is imperative to acknowledge that the mechanisms underlying the interaction between RNA viruses and host cell LDs can vary significantly across different virus types and host cell types. Various viruses utilize LDs to differing degrees and employ diverse strategies to exploit them for their replication and survival. A comprehensive understanding of the molecular interactions and mechanisms governing the virus/LD relationship is paramount for the development of targeted antiviral strategies aimed at disrupting this process and inhibiting viral replication. While it is well established that members of the Flaviviridae family often interact with LDs within infected cells, the relationships between LDs and flaviviruses have been more extensively investigated in viruses such as DENV, ZIKV, and HCV (Table 1). However, it is essential to recognize that the biological effects of LDs on different flaviviruses may vary. Despite some recent studies exploring the connections between LDs and other flaviviruses such as tick-borne encephalitis virus (TBEV) [75], West Nile virus [76], and Japanese encephalitis virus (JEV) [77], many aspects of these interactions remain largely unexplored and require further investigation. Understanding the intricacies of the interplay between LDs and various RNA viruses, including flaviviruses and others, will shed light on novel targets for antiviral intervention and contribute to our broader understanding of virus/host interactions.

Studies examining the interactions between flaviviruses and LDs at the protein level have shown that capsid proteins of all Flaviviridae members are linked to LDs [76,78,79,80]. These associations occur independently of viral activities, as the expression of capsid protein alone is sufficient to trigger the colocalization of capsid with LDs. However, it is noteworthy that the capsid protein alone does not exert significant effects on the biogenesis and breakdown of LDs. This suggests that LDs play a crucial role in the viral replication cycle, serving as a platform for various stages of viral replication and assembly.

### 3.4. DENVs and LDs

#### 3.4.1. DENV Capsid Protein and LDs

DENV has been extensively investigated for its interactions with LDs. Initial findings indicated that the mature capsid protein in the cytoplasm of DENV-infected cells accumulates on the LD surface, and the involvement of two hydrophobic amino acids, L50 and L54, was identified as crucial for this LD association [81]. The interaction was subsequently validated through Atomic Force Microscopy (AFM)-based force spectroscopy measurements, and TIP47 was identified as the LD surface protein responsible for this interaction [78]. Consistently, the DENV capsid protein was found to interact with very low-density lipoproteins (VLDLs), which are LDs present in plasma secreted from adipocytes. Notably, this interaction was not observed with low-density lipoproteins (LDLs), characterized by lower TAG and higher cholesteryl ester (CE) content than VLDLs [82]. Interestingly, the positively charged N-terminal region of the capsid protein was found to interact with the negatively charged LD surface, likely involving TIP47. This interaction led to the development of a peptide with inhibitory activity against C protein/LD binding, offering a potential avenue for the development of new drugs against DENVs [83]. Moreover, a small peptide derived from the C-terminal of the capsid protein demonstrated the ability to penetrate the LD envelope, suggesting a potential interference with DENV’s interaction with LDs [84]. In light of these findings, strategies targeting the capsid/LD protein TIP47 or capsid/LD lipid interactions emerge as promising approaches for the development of innovative anti-DENV treatments.

#### 3.4.2. DENV NS2B3 Protein and LDs

Another documented molecular interaction between the DENV and LDs involves DENV NS2B3 interacting with Rab18, a member of the Rab GTPase family and a crucial regulator of membrane trafficking and FASN, an FA synthetase [85]. This interaction is needed for transporting NS3 to viral replication sites. Rab18 is identified as an LD-associated protein and plays a role in regulating ADRP function, influencing the interplay between LDs and the ER [43]. Experiments involving the depletion of Rab18, the overexpression of Rab18, and the use of either active GTP-bound Rab18 or GDP-bound inactive Rab18 has revealed that Rab18 is supportive of DENV replication and gene expression [85]. Furthermore, Rab18 has been established as a significant cellular factor in the viral assembly process by facilitating the trafficking of the core protein to LDs [86]. Despite these findings, the functional understanding of Rab18 has been limited to a few viruses, and more comprehensive investigations are essential to elucidate whether Rab18 plays a role in various aspects of viral assembly, maturation, and egress. Expanding the scope of functional studies on Rab18 could provide valuable insights into its broader role in the context of viral infections beyond DENV.

#### 3.4.3. DENV-Induced LD Dynamics

While some studies have reported an increase in the numbers and intensities of LDs during DENV infection (Table 1), the mechanisms underlying this LD augmentation remain undisclosed [81,87]. Conversely, other investigations have noted that DENV infection leads to the degradation of LDs through the activation of lipophagy (Table 1). This degradation process can be rescued by autophagy inhibitors such as 3-methyl adenine (3MA) and Etomoxir [88]. Furthermore, it has been proposed that lipophagy-mediated LD degradation is linked to the activation of beta-oxidation, which releases energy necessary for DENV replication. This hypothesis is supported by the observation that autophagy inhibitors, by preventing lipophagy, decrease viral replication [88]. In line with these findings, mechanistic studies have demonstrated that DENV infection-induced lipophagy relies on the activation of AMPK (AMP-activated protein kinase). Knocking down AMPK using siRNA not only hampers LD degradation induced by DENV infection but also attenuates viral replication [89]. Additionally, in the context of DENV-mediated lipophagy, DENV NS4a associates with AUP1, an LD-localized type-III membrane protein, to promote DENV-induced lipophagy, leading to the degradation of LDs in favor of viral replication [90]. It is noteworthy that the observed discrepancies in LD dynamics during DENV infection, including both increases and decreases, may not be contradictory. The increase in LDs was predominantly observed in monocytes and macrophages, whereas the decrease was noted in infected permissive cells. This divergence suggests that the interplay between DENV and LDs may be cell-type specific, highlighting the complexity of the host/virus interaction in different cell populations.

**Table 1 microorganisms-12-00647-t001:** **Viral effects on LDs.** Both RNA and DNA viruses have been shown to influence LDs following infection. These effects can manifest as reduction, increase, or no change in LDs within infected cells. However, conflicting findings exist among studies examining these effects. In the comments section, we carefully analyzed the data presented in the referenced papers and noted that many studies lacked internal controls to compare the viral effects on LDs between infected and uninfected cells under the same experimental conditions. This highlights the need for standardized experimental setups to accurately assess the viral impacts on LDs.

Virus	Cell Line	LD Regulation	Reference	Comments
DENV-2	BHK	Increase	[81]	LD numbers increase
DENV-2, 3	Leukocytes, Macrophage	Increase	[87]	Permissiveness is of question in these cells
DENV-2	Huh-7.5	Decrease	[88,91,92]	Inducing autophagy
DENV-2	HeLa	Decrease	[91]	Inducing autophagy
DENV-2	HEp-G2	Decrease	[89,90]	Inducing autophagy
ZIKV	Placental cells	Increase	[93]	The result shown in the paper is unclear
ZIKV	Huh-7.5	Decrease	[94,95]	
ZIKV	Human Astrocytes	Decrease	[96]	
ZIKV	Human THP-1 monocytes	Increase	[97]	Permissiveness is of question
ZIKV	SH-SY5Y	Increase	[98]	LDs in Mock cells are very low
ZIKV	Neural stem cells	Increase	[98]	LDs in Mock cells are very low
ZIKV	HeLa cells	Decrease	[99]	
HCV	Huh-7.5	Increase	[100,101]	May be related to the viral pathogenesis in hepatic steatosis
PEDV	Vero	Increase	[102]	
Rota virus	Human intestinal enteroids (HIEs)	Increase	[103]	It used a DGAT-/- HIE cell line, no WT HIE cells were used as a control
Poliovirus	HeLa	Decrease	[104]	
RSV	A549, PC-9	Decrease	[105]	
Rabies virus	Neuroblastoma N2a	Increase	[106]	
SARS-CoV-2	Monocyte, A549, Vero E6, HMVEC-L	Increase	[107]	Only Vero E6 is permissive
SARS-CoV-2	293T-hACE2, Caco2	Increase	[108]	DsRNA as viral infection marker
SARS-CoV-2	Vero E6	Increase then decrease	[109]	
SARS-CoV-2	Calu-3	Increase	[110]	
HCMV	Human fibroblast	Increase	[111]	HCMV infection is not clear
KSHV	Tert-immortalized microvascular endothelial (TIME)	Increase	[112]	KSHV infection is not clear
KSHV	HUVEC	Increase	[113]	KSHV infection is not clear
EBV	NP69, HK1	Increase	[114]	Not permissive cell lines
EBV	Nasopharyngeal carcinoma (NPC) cell	Unknown	[115]	

A clinical and in vitro infection study provided evidence that DENV-2/-3 infection induces an increase in the abundance and numbers of LDs in leukocytes and macrophages of Dengue Hemorrhagic Fever (DHF) patients as well as in macrophages infected in vitro with DENV-3 [87]. This study observed higher levels of migration inhibitory factor (MIF) in the plasma of DHF patients, and MIF was found to colocalize with LDs in DENV-infected cells. Considering that LDs store eicosanoids, which are associated with innate immunity and inflammatory responses, it is suggested that the LD-MIF pathway may contribute to DENV-induced consequences in DHF and inflammation. In a related study, the same group demonstrated that DENV infection induces inflammation, particularly through platelet/monocyte interactions. Upon DENV infection, platelets aggregated with monocytes, enhancing LD biogenesis in monocytes. Inhibiting LD formation resulted in the attenuation of prostaglandin E2 (PGE2) secretion [116]. Once again, the DENV-induced increase in LD formation in monocytes appears to be an indirect process mediated by an unknown pathway, possibly associated with DENV-induced inflammation. Further investigations in this area may establish a link between LD and DENV pathogenesis.

Given the crucial role of LDs in DENV replication, targeting LD formation or trafficking could be a viable approach to inhibit DENV replication. One such example is Brefeldin A (BFA), a drug that blocks the transportation of the DENV capsid protein from the Golgi to LDs by interfering with the GBF1-Arf1/Arf4-COPI pathway. This interference results in inhibitory effects on DENV replication [117]. Another compound, PF-429242, acts as an inhibitor of site 1 protease (S1P), which activates sterol regulatory element-binding proteins (SREBPs). SREBPs are crucial for regulating intracellular lipid levels. PF-429242 has been shown to significantly reduce cellular LDs, resulting in more than a 100-fold inhibition of DENV replication [91,92]. Considering the studies on the interactions of the DENV capsid protein with LDs, the hydrophobic domain of the capsid protein emerges as a specific target for the development of anti-DENV drugs. Indeed, a study investigating 18 aromatic derivatives of guanylhydrazones and aromatic oximes, known for their ability to bind to the DENV capsid protein, identified five promising candidates with inhibitory effects on DENV infection and replication [118]. In conclusion, the interaction between DENV and LDs presents an intriguing avenue for the exploration of potential antiviral strategies, with a specific focus on disrupting LD formation or trafficking and targeting the hydrophobic domain of the DENV capsid protein. These findings hold promise for the development of effective anti-DENV drugs.

### 3.5. ZIKV and LDs

ZIKV emerged as a significant viral pathogen during the 2015–2016 epidemic, causing severe health consequences, including microcephaly in neonates born to mothers who were infected with ZIKV during pregnancies [119]. The interaction between ZIKV and LDs was initially explored in a study on ZIKV’s inflammatory effects in the human placenta [93]. ZIKV infection caused impairments of LD biogenesis and intracellular membrane rearrangement, which potentially led to mitochondria dysfunction and placental issues. Although the study showed that ZIKV infection induced the accumulation of LDs in primary placental cells including stromal and trophoblast cells, the effects on LDs in placental cells remain unclear, with conflicting results on ZIKV-induced LD alterations. It is unclear whether the effects of ZIKV on LDs of placental cells are direct or indirect as they showed that a ZIKV cell has less LDs than the uninfected cells.

Similarly, to DENV, it is uncertain whether ZIKV infection leads to a decrease or increase in LDs within infected cells. While reports indicate LD accumulation in primary placental cells and human monocytes [93,97], ZIKV has been shown to decrease LD intensity and numbers in hepatocytes and HeLa cells [94,95]. This raises the question of whether ZIKV’s impact on LD regulation is cell specific.

In contrast to HCV and similar to DENV, ZIKV encodes only one protein, capsid, to localize with LDs [80,120]. The ZIKV capsid protein, when expressed alone by transfection, localizes in LDs, the Golgi apparatus, and nucleoli [80,120]. Interactome mapping by Coyaud et al. [120] revealed that the ZIKV capsid protein interacts with proteins involved in vesicle trafficking, RNA processing, and lipid metabolism. Additionally, the ZIKV capsid protein directly interacts with viral RNA [121]. Given that the ZIKV capsid protein interacts with LDs and viral RNA and that viral replication causes LD reprogramming, we hypothesize that the interaction of LDs and the capsid protein may be crucial for viral particle assembly. Unfortunately, the above-mentioned findings on the ZIKV capsid protein were all generated from the transfection system; a deeper investigation in the context of the infection system is required to demonstrate the capsid/LD interaction for ZIKV.

Autophagy flux was found to be activated by ZIKV infection, triggering the AMPK-mediated activation of Unc-51-like kinase 1 (ULK-1), initiating autophagy in the ER and potentially lipophagy in LDs (Figure 3). The ZIKV-induced lipophagy influences the LD-related lipid metabolism [95]. As autophagy initiation, LD formation, and ZIKV replication occur in the ER, the interplay between the ER and ZIKV is hypothesized to be critical in viral replication. ZIKV activates the unfolded protein response (UPR) pathway in ER stress, often occurring post-viral replication, leading to translation inhibition and protein degradation through sensors like inositol-requiring enzyme 1α (IRE1α), protein kinase R (PKR)-like ER kinase (PERK), and transcription factor 6 (ATF6). ZIKV activates IRE1α to promote X-box binding protein 1 (XBP1) splicing [122], a regulator of UPR-related gene expression. Balancing ZIKV infection and ER function ensures optimal viral replication.

### 3.6. HCV and LDs

#### 3.6.1. HCV Capsid Protein and LDs

HCV targets hepatocytes in the liver, leading to hepatitis and various chronic liver diseases [123]. Notably, hepatocytes are the primary site of HCV infection. Additionally, LDs are notably more prevalent in hepatocytes compared to other cell types, with adipocytes being the exception [14]. The relationship between LDs and HCV has been the most investigated among all the viruses. Initial investigations utilizing immunofluorescence analysis (IFA) and electron microscopy showcased the colocalization of the HCV capsid protein with LDs in both nonhepatic cells (CHO) and hepatic cells (HEp-G2) [124]. This LD association has also been noted with capsid proteins of other flaviviruses such as ZIKV and DENV.

Subsequent inquiries unveiled that the HCV capsid protein is situated on the surface of LDs. Molecular and functional analyses pinpointed the significance of domain 2 (residues 119–174) of the capsid protein p21 in localizing to LDs [125]. This region also proves pivotal for its association with the ER and mitochondria [126]. Structurally, domain 2 comprises two amphipathic alpha helices separated by a hydrophobic loop, with the loop playing a pivotal role in efficient LD association. The hydrophobic loop is required for the helices to fold, and a combination of Helix I, the hydrophobic loop, and Helix II is essential for the HCV capsid protein to have efficient LD association [127]. The importance of the helices in domain 2 extends beyond HCV, encompassing other viral capsid proteins’ association with LDs. (Table 2). A noteworthy discovery involves Rab18, a Ras-related small GTPase, participating in the colocalization of the capsid protein with LDs. The knockdown of Rab18 impedes the recruitment of the HCV capsid protein to LDs, underscoring its significance in this process [86]. Nevertheless, further research is imperative to comprehensively elucidate the interactions of the HCV capsid protein with cellular proteins both before and after HCV infection, shedding light on its intricate role in HCV pathogenesis.

#### 3.6.2. Other HCV Proteins and LDs

In contrast to ZIKV and DENV, where only viral capsid proteins have been found to be related to LDs, HCV encodes multiple proteins associated with LDs. Notably, NS5A, a non-structural protein initially identified as an ER protein, exhibits colocalization with the HCV capsid protein on LDs [131]. This colocalization is independent of other HCV proteins, as the expression of NS5A from a transfected NS5A-expressing plasmid also resulted in its presence on LDs [132]. The interaction of NS5A with TIP47 may be essential for its LD localization, given that TIP47 is an LD-associated protein [133]. The NS5A protein of HCV has been demonstrated to engage with various cellular proteins, playing a role in the modulation of cell growth, interferon resistance, and other cellular signaling pathways. Its interaction with apolipoprotein A1 (apoA1), one of the components of high-density lipoprotein (HDL) particles, led to the discovery of NS5A with LDs. Following viral entry and uncoating in the cytoplasm, the initiation of viral gene expression and replication occurs on ER-derived modified membranes. Although the viral assembly site is not definitively established, it is believed to involve LDs. NS5A’s interaction with TBC1D20 and its cognate GTPase Rab1 is essential for the localization of these proteins to LDs after HCV infection. As Rab1 is necessary for NS5A to be present in viral replication sites, and NS5A is indispensable for HCV replication, the localization of NS5A to LDs and its interaction with TBC1D20 and Rab1 may activate the metabolism of LDs to promote the viral life cycle [132]. Given NS5A’s interaction with LDs and LD-associated proteins, it represents a promising target for antiviral development.

Another HCV protein associated with LDs is NS4B, which possesses multiple biological functions, including inducing the membranous replication platform, RNA binding, NTPase activity, and genetic interaction with NS5A [79]. Unlike NS5A, which either colocalizes with or surrounds the core of LDs, NS4B was observed to be localized at the margins of LDs [134]. The presence of multiple HCV proteins associated with LDs is not unexpected, especially if LDs play a role in the assembly of HCV particles. It is crucial to note that unlike ZIKV and DENV infections, which disrupt or eliminate LDs, HCV infection has not been reported to decrease LDs (Table 1). This distinction may be attributed to HCV’s specific infection of hepatocytes, which naturally contain abundant LDs, exceeding the requirement for viral replication and assembly. Moreover, hepatocytes undergo rapid metabolism, leading to the accelerated generation of LDs.

## 4. Other Viruses and LDS

### 4.1. Other RNA Viruses and LDs

Functional and biological relationships with LDs have also been reported for other RNA viruses. Several RNA viruses exhibit intricate relationships with LDs, implicating the significance of LDs in viral infection and replication. Noteworthy examples include Rota virus, poliovirus, respiratory syncytial virus (RSV), Rabies virus, and SARS-CoV-2.

(1) Rota virus, a prominent pathogen causing acute gastroenteritis in children worldwide, belongs to the Reoviridae family. It encodes various proteins such as NSP2, NSP5, VP1, VP2, and VP6, which interact with LD-related PAT family proteins in the viroplasm. Studies have shown that while Rota virus infection induces the accumulation of PAT family proteins, the overall levels remain unchanged [135]. However, LDs appear crucial for Rota virus replication, as evidenced by the inhibition of replication upon blocking LD synthesis pathways [136]. Intriguingly, Rota virus infection also leads to the degradation of DGAT1, an enzyme required for TAG synthesis and LD formation, raising questions about the regulation and biological role of LDs in Rota viral infection [103]. Therefore, it remains unclear whether Rota virus infection regulates the numbers and abundance of LDs in cells. Moreover, the biological role of LDs in Rota viral infection and replication still needs to be established.

(2) Poliovirus infection in HeLa cells disperses LDs by hydrolyzing neutral lipids stored within them. This process provides activated phospholipids for extensive membrane remodeling in infected cells, aiding in the formation of complex membranous replication structures and protecting the virus from the cellular antiviral response [104].

(3) Respiratory syncytial virus (RSV) disperses LDs in the infected human lung epithelial cells, as demonstrated by Dai et al. [105]. Their study revealed that the dispersion of LDs by RSV correlates with an increase in medium- and long-chain FAs, including myristic acid (14:0), palmitic acid (16:0), oleic acid (18:1), and linoleic acid (18:2). Additionally, a reduction in PPARγ expression and an elevation in the FA catabolism gene PPARα were observed. Consequently, this cascade of events leads to lipid oxidation, the release of free FAs, and an upregulation of pro-inflammatory cytokines such as IL-1, IL-2, IL-4, and IL-6. These inflammatory responses contribute to increased inflammation and oxidative damage within lung tissues. Thus, the interaction between RSV and LDs may play a pivotal role in RSV pathogenesis within the respiratory system.

In contrast to RSV, the rabies virus has been observed to elevate both the cellular intensity and quantity of LDs in infected mice and cells [106]. Interestingly, reducing LDs using compounds like atorvastatin resulted in the inhibition of viral production. However, the study requires more robust controls to draw definitive conclusions. Notably, LDs were not visualized in non-infected cells during the rabies virus infection assays, while LDs were clearly observed in the same cell line when cells were treated with atorvastatin. This discrepancy underscores the need for further investigation to establish a clearer understanding of the relationship between rabies virus infection and LD dynamics [106].

(4) SARS-CoV-2, the pathogen of COVID-19, has been linked to LDs. Initially, studies on primary monocytes isolated from COVID-19 patients revealed a higher LD content compared to those from non-infected individuals [107]. Subsequent studies using cell lines, including Vero E6, A549, and HMVEC-L, consistently, showed increased LD levels upon SARS-CoV-2 infection. This elevation in LDs may stem from enhanced LD synthesis, supported by the upregulation of proteins such as SREBP-1, DGAT-1, CD36, and PPAR-γ following SARS-CoV-2 infection [107]. However, it is noteworthy that A549, monocytes, and HMVEC-L cells are not permissive to SARS-CoV-2 infection, and the absence of viral protein detection raises concerns regarding the specificity of these findings. In contrast, recent research conducted on Vero E6 cells demonstrated elevated levels of pHSL, an activated lipase, and an increase in PLIN3, a protective protein of LDs, during the early stages of SARS-CoV-2 infection, followed by a decrease later on [109]. Similar to HCV-infected cells, multiple studies suggest that SARS-CoV-2 replication may occur within or in association with LDs [107,137]. The presence of viral components such as dsRNA, N protein, and NSP6 protein within LDs supports the proposed role of LDs in viral replication, maturation, and assembly [138,139].

However, the dynamics of SARS-CoV-2-induced LDs require further clarification. Although it was stated that SARS-CoV-2 induced LD accumulation overtime, some studies indicate a decrease in LD intensity in cells with higher levels of viral dsRNA, suggesting a nuanced relationship [138]. In search of the viral proteins that may be responsible for SARS-CoV-2 to induce LD accumulation, the effects of global SARS-CoV-2 proteins on LD numbers in cells were screened; only ORF3a was found to be able to increase the numbers of LD [140]. However, a mutagenesis study showed that NSP6 is the viral protein that is critical for SARS-CoV-2 to induce LDs [141]. In conclusion, understanding the interaction between SARS-CoV-2 and cellular LDs is crucial, as LD levels may directly impact the inflammatory response induced by the virus. Further research is warranted to elucidate the intricate mechanisms underlying SARS-CoV-2-mediated alterations in LD dynamics and their implications for viral pathogenesis and host immune responses.

### 4.2. DNA Viruses and LDs

Investigations into the effects of LDs on DNA viruses have been less extensive compared to those on RNA viruses. Nonetheless, several DNA viruses, primarily human herpesviruses (HHVs), including Kaposi’s Sarcoma-associated herpesvirus (KSHV), Human cytomegalovirus (HCMV), herpes simplex virus 1 (HSV-1), and Epstein–Barr virus (EBV), have been studied for their biological relationships with LDs and LD-associated proteins.

(1) HCMV: HCMV infection triggers LD responses dependent on two lipid metabolism proteins: Viperin and SREBP. Viperin (Virus inhibitory protein, endoplasmic reticulum-associated, interferon-inducible), encoded by the (Radical S-adenosyl methionine domain-containing protein 2) gene, is an interferon-inducible antiviral protein with multiple functions [142]. It is an ER protein and localizes in LDs and defends against viral replication [143]. HCMV induces the translocation of Viperin to mitochondria, inhibiting the function of the mitochondrial trifunctional protein (TFP) that mediates fatty acid β-oxidation. This subsequently causes an increased expression of lipogenesis genes, particularly sterol regulatory element-binding proteins (SREBPs) and carbohydrate-responsive element-binding protein (ChREBP), resulting in LD accumulation [144]. Although HSV-1 gD protein has similar effects on Viperin after transfection [145], viral effects on LDs have not been extensively reported.

(2) KSHV: KSHV is an oncogenic virus associated with cancerous diseases in immune-compromised populations. KSHV infection induces neural lipid biogenesis including TAG and CE and LD accumulation during both latent and lytic infection phases [112,113]. However, conclusive information regarding the lytic phase of KSHV infection is required. Lipid synthesis appears to be proviral, as FA synthesis inhibitors selectively induce apoptosis in latently infected KSHV cells.

(3) EBV: EBV interacts with LDs, especially during the latent stage of infection. The EBV latent protein LMP1 plays a critical role in promoting the expression of FA synthesis genes, including SREBP [114] and FASN [146]. Furthermore, EBV-encoded protein LMP2A downregulates ATGL inhibiting lipid degradation [115]. These mechanistic pathways promoting LD accumulation may apply to KSHV, given the genetic and pathogenic similarities between KSHV and EBV.

In summary, while research on the interactions between LDs and DNA viruses, particularly HHVs, is relatively limited, studies suggest a significant involvement of LDs in the life cycle and pathogenesis of these viruses. Further investigations are warranted to fully elucidate the mechanisms underlying LD/virus interactions and their implications for viral replication and disease progression.

## 5. LD Metabolism and Its Impact on Viral Replication

LDs typically maintain a homeostatic state during cellular quiescence; however, this does not imply that LDs remain static. LD biogenesis and breakdown are ongoing processes essential for normal cellular life cycles. Upon viral infection, viruses hijack functional organelles for their own replication needs. While LD biogenesis may still be necessary, the breakdown of LDs needs to be controlled by viruses. In a cell culture system, the initial response of permissive cells to viral infection involves the shutdown of cellular activities, including protein synthesis, DNA replication, cell division, and cell migration [147]. In addition to maintaining the infected cell viability, the virus must co-opt cellular functions for viral replication, with LDs playing a crucial role.

LDs are integral to viral replication, although the specific interactions between LDs and viruses can vary. Utilizing LDs for viral replication appears to be a common strategy across different viruses. As depicted in Figure 5, the degradation of LD accompanies the catabolism of neutral lipids such as TAG and CE, generating FA and CE and SM and PC. FAs are translocated into the mitochondrion after chemical modification by enzymes such as carnitine palmitoyltransferase 1 (CPT1), carnitine acyltransferase (CAT), and CPT2. Within the mitochondrion, FAs undergo beta-oxidation to generate abundant ATP, which is critical for viral gene expression, RNA replication, protein synthesis, and overall viral replication. Cholesterol esters, sphingomyelin, and phosphatidylcholine are important for the formation of viral replication compartments and viral assembly. The explanations for the diverse consequences of LDs upon viral infection can be understood through two layers: (1) certain viruses like ZIKV and DENV rapidly deplete LDs upon entering cells to generate ATP for various viral processes, including RNA replication, protein translation, and viral maturation, assembly, and egress; and (2) conversely, viruses such as HCV may require the accumulation of LDs post-infection but do not cease the degradation of LDs for viral use. However, HCV may promote more LD biogenesis than LD degradation as part of its replication strategy.

The interaction between viruses and LDs likely involves a level of complexity beyond our current understanding. LDs typically form within the leaflets of the ER bilayer as oil droplets or “lenses” that eventually bud towards and are released into the cytoplasm. However, LDs are not isolated entities; they establish connections, both direct and indirect, with other organelles through LD contacts. These contacts, including LD-ER, LD-mitochondria, LD/lysosome, and LD-LD interactions, involve proteins located on the surfaces of the interacting organelles [68,69,70,148,149].

The biological functions of LD contacts have not been fully elucidated, but hypotheses suggest their involvement in various cellular processes. For instance, LD/mitochondria contacts may be linked to FA metabolism within mitochondria, providing energy for cellular activities. LD/lysosome contacts might aid in the removal of excess lipids, safeguarding cells from potential toxicity. LD-ER contacts are associated with LD maturation and are crucial for dynamic cellular responses to environmental cues. Additionally, LD-LD contacts may facilitate the fusion of LDs, potentially optimizing lipid storage efficiency by forming larger droplets. Further exploration is needed to unravel the precise roles of LD contacts and their impact on cellular physiology and viral infections. Investigating these interactions could provide valuable insights into cellular dynamics and unveil potential targets for therapeutic intervention in viral diseases.

Interestingly, all the features of LD contacts mentioned above could be reasonably applied to virus-infected cells. LD/mitochondria contacts may play roles in providing energy for viral gene expression, replication, assembly, and egress. LD-ER contacts may be important for the virus to establish replication complexes alongside the ER. LD/lysosome contacts are crucial for transporting mature viral particles for egress. Lastly, LD-LD contacts may be necessary for viral replication outside LDs, as some viruses have been observed replicating in or alongside LDs. However, there is little evidence to support these notions regarding the biological functions of LD contacts. The knockdown or knockout of genes essential for LD contacts would be a powerful tool not only to investigate the functions of LD contacts but also to explore their effects on viral replication.

## 6. Perspective of LDs in Antiviral Strategies

Viral diseases constitute a significant portion of global infectious diseases and are responsible for a substantial number of annual deaths worldwide. According to estimates by the World Health Organization (WHO), viral diseases contribute to approximately one-third of the total global deaths each year [150]. Current efforts in antiviral development have primarily focused on targeting viral processes or enzymes. This strategy aims to achieve potent antiviral effects while minimizing host cell toxicity by avoiding the cross-inhibition of host proteins [151]. Despite the progress made in antiviral drug development, the lack of a wide-spectrum drug against diverse viruses remains a challenge. The emergence and re-emergence of viral infections can occur unpredictably, necessitating the urgent development of broad-range antivirals. To address this, a new approach focuses on targeting host cell pathways and enzymes that are essential for virus replication. Viruses depend on the host cell environment for their replication, making virus/host interactions crucial during the disease process. By disrupting these virus-dependent host cell pathways and enzymes, broad-spectrum antivirals can effectively inhibit viral replication across different viral strains. This paradigm shift offers promising opportunities for the development of effective antivirals capable of combating a wide range of viral infections.

Among the host targets, LDs emerge as a potential target of choice. Regardless of the viral effects on LDs in the infected cells, reducing LDs or inhibiting LD biogenesis proves inhibitory to viral replication. This is supported by the essential role LDs play in viral gene expression, replication, and assembly. Additionally, LDs contribute to inflammation, a common pathogenic effect of viral diseases. Thus, reducing LDs not only attenuates viral infection and replication but also decreases virus-induced inflammation. Moreover, LDs are necessary for malignant cell growth, suggesting that anti-LD strategies could also be applied to antineoplastic treatment. In summary, lipid metabolism presents a novel and practical source of potential targets for antiviral discovery against different viruses, offering the promise of developing antivirals effective against a broad spectrum of viruses.

## Figures and Tables

**Figure 1 microorganisms-12-00647-f001:**
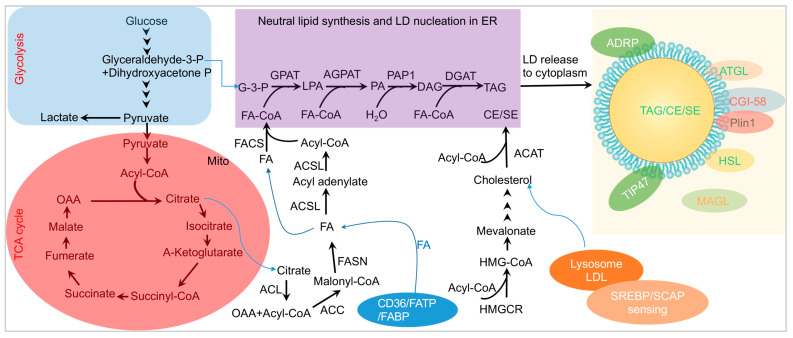
A schematic model of LD biogenesis. FA, serving as the material for TAG synthesis in the ER, is acquired through the CD36-mediated pathway or produced from Acetyl-CoA by ACC and FASN. Acetyl-CoA is generated from citrate, a product of glucose breakdown in the mitochondrion. FA-CoA and G-3-P (a product from glycolysis) participate in the de novo synthesis of TAG in the ER. CE is formed from cholesterol, a component of LDL sensed by SREBP/SCAP, and Acyl-CoA. Finally, LDs are biogenesized and released to the cytosol and surrounded by various cellular proteins. **Abbreviations.** LDs, lipid droplets; ER, endoplasmic reticulum; Mito, mitochondria; FA, fatty acid; ACSL, long-chain-fatty-acid—CoA ligase; Acyl-CoA, acyl coenzyme A; G-3-P, glycerol-3-phosphate; GPAT, glycerol-3 phosphate acyltransferase; LPA, lysophosphatidic acid; AGPAT, acylglycerolphosphate acyltransferase; PA, phosphatidic acid; PAP, phosphatidic acid phosphatase; DAG, diacylglycerol; DGAT, acyl-CoA:diacylglycerol acyltransferase; TAG, triacylglycerol; Plin, perilipin; LDL, low-density lipoprotein; CE, cholesterol ester; ATGL, adipocyte triglyceride lipase; MAGL, monoacylglycerol lipase; HSL, hormone-sensitive lipase; CGI-58, comparative gene identification-58; SREBP, sterol regulatory element-binding protein; SCAP, SREBP1 cleavage-activating protein; ACC, Acetyl-CoA Carboxylase; FASN, fatty acid synthase; TIP47, Tail-interacting protein 47 (also called Plin3), TCA, tricarboxylic acid; OAA, oxyloacetic acid; P, phosphate; ACL, ATP citrate lyase; ACC, acetyl-CoA carboxylase; FAS, fatty acid synthase; HMG, 3-hydroxy-3-methylglutaryl; HMGCR, HMG-CoA reductase; FACS, fatty acyl-CoA synthetase; FATP, fatty acid transport protein; FABP, fatty acid-binding protein.

**Figure 2 microorganisms-12-00647-f002:**
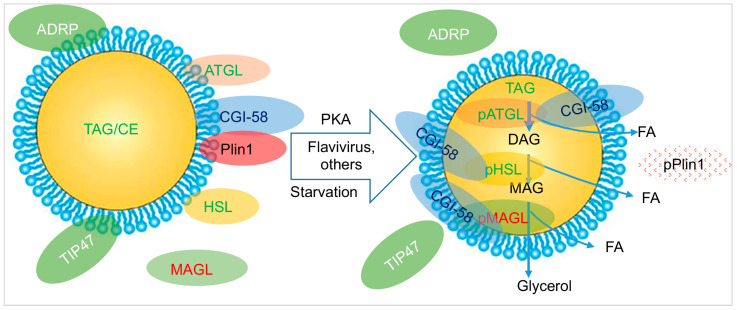
**Lipolysis of LDs:** Perilipin 1 (Plin1) binds with CGI-58 to keep it away from ATGL, HSL, and MAGL so that the lipases are at an inactive state and LDs maintain intact (left); several conditions such as starvation, viral infection, the demanding of membrane generation, and stress activate PKA that phosphorylates Plin1, leading its degradation (right). In this case, CGI-58 is freed to bind ATGL, HSL, and MAGL causing their phosphorylation and activation to enter the LD core to lysate TAG and DAG to release FAs and from MAG that is catalyzed to produce glycerol and FAs. **Abbreviations**. pPlin, phosphorylated Plin; PKA, protein kinase A.

**Figure 3 microorganisms-12-00647-f003:**
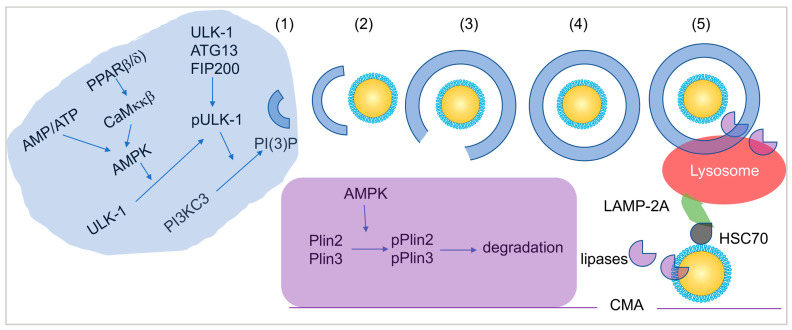
Lipophagy of LDs. Upper panel. Macrophagy of LD lipophagy: (1) Preautophagosome assembly, initiation site of phagosome formation with molecular activation cascade (blue box). (2) phagophore membrane expansion to surround LDs, (3) vesicle elongation, (4) vesicle nucleation to form closed autophagosome, and (5) fusion of matured autophagosomes with lysosome. Lower panel. Chaperone-mediated autophagy depends on interactions of LAMP2A and HSC70, resulting in degradation of Plin2 and Plin3 (purple box) and LDs. **Abbreviations**. AMPK, AMP-activated protein kinase; AMP, adenosine monophosphate; ATP, adenosine triphosphate; PPAR, peroxisome proliferator-activated receptor; CaMKKβ, Ca(2+)/CaM-dependent protein kinase kinase β; ULK-1, Unc-51-like kinase 1; PI3KC3, phosphatidylinositol (PI)3-kinase complexes 3; PI(3)P, Phosphatidylinositol 3-phosphate; LAMP-2A, lysosome protein 2A; CMA, chaperone-mediated autophagy; HSC70, Heat shock cognate 71 kDa protein.

**Figure 4 microorganisms-12-00647-f004:**
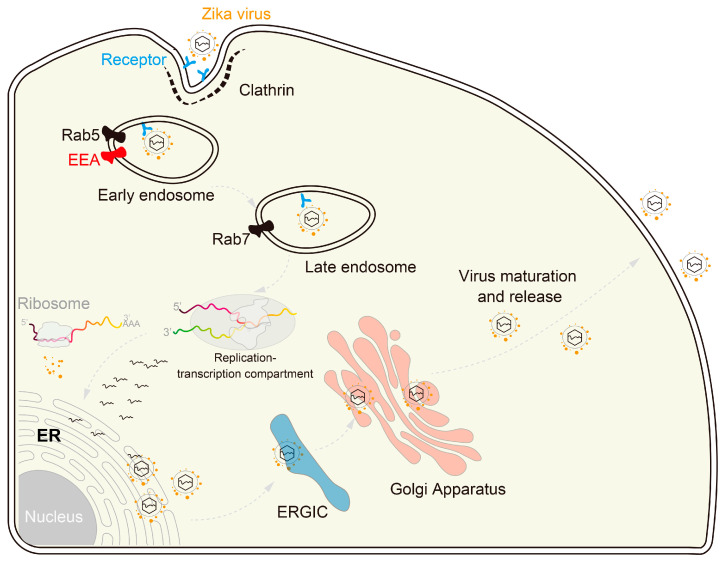
Zika virus replication cycle. ZIKV entry, trafficking to ER for viral replication, assembly, and maturation in Golgi apparatus and release outside of cells.

**Figure 5 microorganisms-12-00647-f005:**
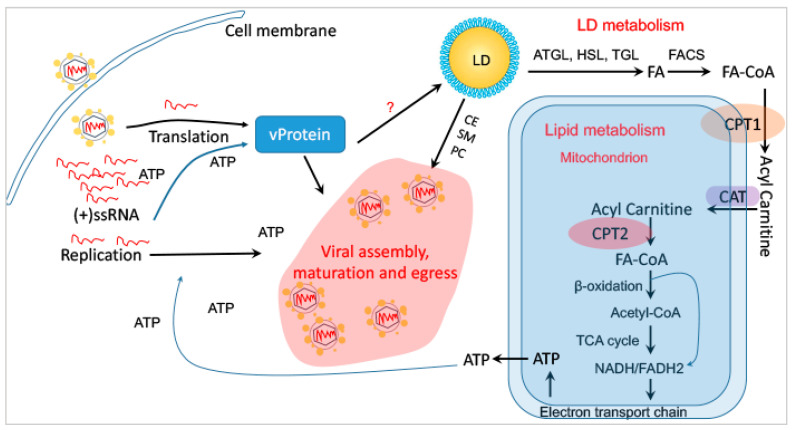
LD metabolism for viral replication within the infected cells. Upon viral infection into a permissive cell, unknown (?) viral effects and factors reprogram LDs to provide necessary materials for viral replication: (1) FA for generating ATP in the mitochondrion and (2) CE/SM/PC for establishing viral replication compartment and viral assembly. **Abbreviations**. ATP, adenosine triphosphate; FASN or FAS, fatty acid synthase; CPT1, carnitine palmitoyltransferase 1; FACS, fatty acyl-CoA synthase; CAT, carnitine translocase; PPL, phospholipid; SM, sphingomyelin; PC, phosphatidylcholine; vProtein, viral protein.

**Table 2 microorganisms-12-00647-t002:** Peptide domains of capsid proteins and LDs. For DENV, ZIKV, and HCV, capsid proteins closely relate to LDs presenting as colocalization. Alpha helix structures in N-terminus of capsid proteins of DENV and ZIKV and in middle of HCV capsid protein are required and sufficient for protein to colocalize with LDs.

Virus	Peptide	References
DENV	Peptide aa14–aa23 (NMLKRARNRV) Peptide aa5–aa26 (RKKTGRPSFNMLKRARNRVSTV)	[83]
ZIKV	Pre-α1 loop peptide 25–36 (SPFGGLKRLPAG)	[121,128,129,130]
HCV	Domain 2 (aa119–aa174)	[125]

## Data Availability

The original contributions presented in the study are included in the article, further inquiries can be directed to the corresponding author.

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
