# Peer review of "Lipid Droplets: Formation, Degradation, and Their Role in Cellular Responses to Flavivirus Infections"

_microorganisms, 2024, doi:10.3390/microorganisms12040647_

Round 1

Reviewer 1 Report

Comments and Suggestions for Authors

Summary: In this review, Hsia et al describe the life cycle of lipid droplets and their importance for flavivirus infection. They extensively cover formation, regulation, and degradation of these organelles and apply this knowledge to flavivirus infection. Overall, the information content is sufficient for the scope of this review, however there are significant issues with organization, clarity, and references throughout the manuscript.

Major comments:

·         Organization of the subsections is not consistent, please try to make writing and subsection organization clear and consistent throughout the review.

·         Reference 3 is the 2023 taxonomic update for negative-sense RNA viruses, flaviviruses are positive-sense RNA viruses.

·         Section 3.3 has a concerning lack of references throughout.

·         Table 1 has major formatting inconsistencies, missing references, and incorrect information

·         Line 466 the 2015-2016 ZIKV outbreak was considered an epidemic by the WHO, not a pandemic

·         Abbreviations are inconsistent throughout the manuscript and often defined multiple times.

Minor comments:

·         The authors should consider labelling figure 1 with the bullets from lines 95-104 in the text to make it a little easier to follow as the figure is busy

·         Line 89 is redundant as written

·         Line 90 “pink box of figure 1” should be changed to pink oval/circle.

·         Line 95 the Kennedy pathway appears to be in a purple box, not gray

·         Line 151-154 Section 2.1.2 number 3, are there any references for the Lipin subsection?

·         Line 172 envelop should be envelope

·         Lines 175-185 in section 2.1.3. the authors should consider changing the numbering format for the list of LDRP categories, as written they look like reference numbers.

·         Lines 205-211 Figure 2 legend is confusing as written

·         Figure 2 is a little confusing to look at—especially the right side of the figure, could the authors please adjust to make this easier to read?

·         Lines 232-237 Could the authors please number the stages of lipophagy in the main text to accompany figure 3?

·         Section 3.1 wording is confusing as written

·         Figure 4 endosome and transcription are misspelled

·         Section 3.3 subsections (LD association etc) and 4.1 should be numbered to be consistent with the rest of the manuscript.

·         Line 528-532 sentence is repeated

·         Lines 649-660 is confusing as written

Comments on the Quality of English Language

English and grammar quality throughout the manuscript are fine.

Author Response

Reviewer 1:

Major comments:

  • Organization of the subsections is not consistent, please try to make writing and subsection organization clear and consistent throughout the review.

Responses:

The manuscript has been reorganized for consistency, and we trust that it is now clear. We extend our gratitude to the reviewer for providing valuable comments and suggestions, which have played a crucial role in enhancing the quality of this manuscript.

  • Reference 3 is the 2023 taxonomic update for negative-sense RNA viruses, flaviviruses are positive-sense RNA viruses.

Responses:The reference has been changed into “Simmonds P, Becher P, Bukh J, Gould EA, Meyers G, Monath T, Muerhoff S, Pletnev A, Rico-Hesse R, Smith DB, Stapleton JT, Ictv Report C. ICTV Virus Taxonomy Profile: Flaviviridae. J Gen Virol. 2017;98(1):2-3. Epub 2017/02/22. doi: 10.1099/jgv.0.000672. PubMed PMID: 28218572; PMCID: PMC5370391.”

  • Section 3.3 has a concerning lack of references throughout.

Responses:The references have been added.

  • Table 1 has major formatting inconsistencies, missing references, and incorrect information

Responses:Table 1 has been revised by adding relevant references and correct information.

  • Line 466 the 2015-2016 ZIKV outbreak was considered an epidemic by the WHO, not a pandemic

 Responses: It has been changed into “epidemic”.

  • Abbreviations are inconsistent throughout the manuscript and often defined multiple times.

Responses:The manuscript has undergone a comprehensive revision of abbreviations to ensure consistency and eliminate redundancy throughout.

Minor comments:

  • The authors should consider labelling figure 1 with the bullets from lines 95-104 in the text to make it a little easier to follow as the figure is busy

Responses:Bullets have been added to the contents.

  • Line 89 is redundant as written

Responses:It has been revised.

  • Line 90 “pink box of figure 1” should be changed to pink oval/circle.

Responses:It has been changed into “pink oval”.

  • Line 95 the Kennedy pathway appears to be in a purple box, not gray

Responses:It has been changed into “purple box”.

  • Line 151-154 Section 2.1.2 number 3, are there any references for the Lipin subsection?

Responses:References have been added.

  • Line 172 envelop should be envelope

Responses:corrected.

  • Lines 175-185 in section 2.1.3. the authors should consider changing the numbering format for the list of LDRP categories, as written they look like reference numbers.

Responses:Thanks a lot for the suggestion, it has been changed.

  • Lines 205-211 Figure 2 legend is confusing as written

Responses:The legend has been revised.

  • Figure 2 is a little confusing to look at—especially the right side of the figure, could the authors please adjust to make this easier to read?

Responses:The figure 2 has been modified.

  • Lines 232-237 Could the authors please number the stages of lipophagy in the main text to accompany figure 3?

Responses:The numbers have been added in the text.

  • Section 3.1 wording is confusing as written

Responses:The section have been revised.

  • Figure 4 endosome and transcription are misspelled

 Responses: The figure 4 has been corrected.

  • Section 3.3 subsections (LD association etc) and 4.1 should be numbered to be consistent with the rest of the manuscript.

Responses:it has been revised with numbering.

  • Line 528-532 sentence is repeated

 Responses: Corrected.

  • Lines 649-660 is confusing as written

Responses:It has been revised.

Reviewer 2 Report

Comments and Suggestions for Authors

After reviewing the manuscript: "Lipid Droplets: Formation, Degradation, and Their Role in Cellular Responses to Flavivirus Infections," I must say it provides a comprehensive overview of the subject. This work is a remarkable and significant contribution that aids in comprehending the mysterious lipid droplets (LDs) and their potential importance in the realm of flavivirus infections. The changes above aim to enhance the organization of the manuscript for improved scholarly clarity.

Recent Literature: It is advisable to include reviews of recent scholarly articles to enhance this work and stay updated on the latest developments in the field. This will not just back up the current findings but could also introduce fresh ideas for discussion within the context of your work.

Explanation of Technical Terms: While the manuscript contains numerous technical details, certain sections may benefit from further clarification of technical terminology, or put simply, simplification. Having glossaries or footnotes to explain difficult terms would greatly benefit readers, especially those unfamiliar with them.

Enhancing the Quantitative Analysis: Delving deeper into the quantitative aspect related to flavivirus replication and LDs could bolster your arguments. You may have been aiming to find comparable outcomes, which would be commendable. Consider conducting either a meta-analysis or a systematic review of these existing studies to provide a comprehensive overview of the evidence.

Enhancing the clarity and resolution of figures and tables to align with the quality of the supporting text would be beneficial. Quality visual aids are essential for helping readers grasp complex biological processes.

Novelty and Implications Expanded: The above-presented details that regard multiple viruses and their targeting of other organelles point to the fact that LDs would represent a novel target for antiviral strategies. The conversation can highlight the importance of the research and the potential real-world implications that the findings could have in the specific field.

It is important to clearly outline any potential limitations and biases that may exist in your research to ensure a balanced and credible review. Transparency is essential for scholarly discussions. Your manuscript offers a valuable perspective on LDs and their interaction with flavivirus infections, pointing towards a promising path for the future of antiviral research.

I am looking forward to your revision and am confident that with the suggested changes, this work could be a valuable contribution to the scientific community.

Comments on the Quality of English Language

The quality of English in the present manuscript is commendable and with a high proficiency level suitable for academic audiences. The paper could, however, be subjected to minor editing for the manuscript to fully serve in scholarly communication. Notably, the changes made were to optimize reading flow through structuring sentences and the terms used technologies to be accurate and definable with clarity. This clarification will help tune up the clarity with which the scientific content reaches its target audience in the manuscript.

Author Response

Reviewer 2.

Recent Literature: It is advisable to include reviews of recent scholarly articles to enhance this work and stay updated on the latest developments in the field. This will not just back up the current findings but could also introduce fresh ideas for discussion within the context of your work.

Responses:We totally agree with the reviewer that review of recent articles can enhance our research. Some more recent relevant references have been cited.

Explanation of Technical Terms: While the manuscript contains numerous technical details, certain sections may benefit from further clarification of technical terminology, or put simply, simplification. Having glossaries or footnotes to explain difficult terms would greatly benefit readers, especially those unfamiliar with them.

Responses:Five figures have been included to elucidate LD biogenesis, breakdown, viral replication cycle, and viral interaction with LDs within infected cells. The terminology employed in this manuscript aligns with standard usage within the field. We adhered to the journal's guidelines for organizing the writing. Additionally, abbreviations have been provided in the figure legends for clarity.

Enhancing the Quantitative Analysis: Delving deeper into the quantitative aspect related to flavivirus replication and LDs could bolster your arguments. You may have been aiming to find comparable outcomes, which would be commendable. Consider conducting either a meta-analysis or a systematic review of these existing studies to provide a comprehensive overview of the evidence.

Responses:We value the reviewer's suggestion regarding the potential use of a meta-analysis or systematic review to offer a comprehensive overview of existing studies. Recognizing that interactions between LD and viruses vary, we will consider this suggestion for our future research endeavors.

Enhancing the clarity and resolution of figures and tables to align with the quality of the supporting text would be beneficial. Quality visual aids are essential for helping readers grasp complex biological processes.

Responses:We agree with the reviewer, and the table 1 and figures 2 and 4 have been revised to enhance the clarity and quality.

Novelty and Implications Expanded: The above-presented details that regard multiple viruses and their targeting of other organelles point to the fact that LDs would represent a novel target for antiviral strategies. The conversation can highlight the importance of the research and the potential real-world implications that the findings could have in the specific field.

Responses:We are at the initial stages of targeting the interaction between lipid droplets and viruses for the development of antivirals. As discussed in the section 6, this research direction aligns with our current focus.

It is important to clearly outline any potential limitations and biases that may exist in your research to ensure a balanced and credible review. Transparency is essential for scholarly discussions. Your manuscript offers a valuable perspective on LDs and their interaction with flavivirus infections, pointing towards a promising path for the future of antiviral research.

Responses:We sincerely thank the reviewer for the positive feedback on our manuscript. We greatly appreciate the constructive comments and suggestions, which have undoubtedly contributed to the improvement of our work.

Round 2

Reviewer 1 Report

Comments and Suggestions for Authors

The manuscript has improved greatly since the first submission, and modifications to writing clarity and organization are appreciated. No major comments for the revised version. Only minor suggestion is for the authors to ensure capitalization is consistent in tables 1 and 2. Capitalization of column titles for both tables is necessary and entries in Table 1 Cell line and LD regulation columns are inconsistent (ex. some cells in LD reg read Increase, others increase).

Author Response

Apologies for the misunderstanding during our work on revision 1. Once again, we extend our gratitude to the reviewer for the meticulous reading and insightful feedback. As per your suggestion, we have capitalized the initial letters to ensure consistency throughout the tables.